# Integrated Metabolomic and Transcriptomic Analysis Reveals the Regulatory Effects of Curcumin on Bovine Ovarian Granulosa Cells

**DOI:** 10.3390/ijms26146713

**Published:** 2025-07-12

**Authors:** Bingfei Zhang, Le Chen, Liping Mei, Xianbo Jia, Shiyi Chen, Jie Wang, Hengwei Yu, Songjia Lai, Wenqiang Sun

**Affiliations:** 1State Key Laboratory of Swine and Poultry Breeding Industry, College of Animal Science and Technology, Sichuan Agricultural University, Chengdu 611130, China; zhangbingfei07@163.com (B.Z.); 15990973537@163.com (L.C.); 15700312879@163.com (L.M.); jaxb369@sicau.edu.cn (X.J.); chensysau@163.com (S.C.); wjie68@163.com (J.W.); 18792427097@163.com (H.Y.); laisj5794@sicau.edu.cn (S.L.); 2Key Laboratory of Livestock and Poultry Multi-Omics, Ministry of Agriculture and Rural Affairs, College of Animal Science and Technology, Sichuan Agricultural University, Chengdu 611130, China; 3Farm Animal Genetic Resources Exploration and Innovation Key Laboratory of Sichuan Province, Sichuan Agricultural University, Chengdu 611130, China

**Keywords:** curcumin, transcriptomic, metabolomics, cattle, granulosa cells

## Abstract

Curcumin is a natural polyphenolic compound known to alleviate follicular developmental abnormalities associated with ovarian dysfunction. However, its precise molecular mechanisms remain to be fully elucidated. In this study, we systematically investigated the regulatory effects of curcumin on bovine ovarian granulosa cells through integrated transcriptomic and metabolomic analyses. A total of 503 and 200 significantly altered metabolites were identified in the positive and negative ion modes, respectively, with enrichment in key pathways such as glutathione metabolism, fatty acid biosynthesis, and the phosphatidylinositol signaling pathway. Transcriptomic profiling revealed 1168 differentially expressed genes (582 upregulated and 586 downregulated) which were significantly enriched in pathways related to glutathione metabolism and cellular senescence. Joint multi-omics analysis further demonstrated that curcumin significantly influenced pathways related to glutathione metabolism, cysteine, and methionine metabolism, as well as multiple forms of programmed cell death, including apoptosis, necroptosis, and ferroptosis. Collectively, these findings suggest that curcumin may enhance the antioxidant capacity and survival of granulosa cells by maintaining redox homeostasis and modulating cell fate. This work provides new insights into the potential cellular mechanisms underlying the protective effects of curcumin on granulosa cell function.

## 1. Introduction

The ovary is a highly organized and complex organ consisting of germ cells (oocytes) and somatic cells, including granulosa cells, theca cells, and stromal cells [1]. Notably, the intimate interactions between granulosa cells and oocytes are essential for determining oocyte quality [2,3]. Granulosa cells facilitate oocyte development by supplying nutrients and metabolites through gap junction-mediated communication [4], and protect oocytes from oxidative stress through intrinsic antioxidant mechanisms [5,6], thereby ensuring proper maturation and ovulation [7]. Apoptosis and autophagy of granulosa cells are physiological processes during follicular development that help eliminate dysfunctional follicles and maintain a pool of healthy ones within the ovary [8]. However, excessive activation of these cell death pathways may lead to follicular atresia and depletion, ultimately contributing to ovarian dysfunction [9].

Curcumin is a natural lipophilic polyphenolic compound; it includes two aromatic ring systems containing o-methoxy phenolic groups, connected by a seven-carbon linker that consists of an α,β-unsaturated β-diketone moiety. This distinctive structure confers strong membrane permeability, allowing efficient cellular uptake [10]. Curcumin has been widely recognized for its extensive biological activities and pharmacological effects, including anti-inflammatory, antioxidant, and anticancer properties, making it a promising therapeutic candidate for diverse health conditions [11,12]. The conjugated double bonds in its structure confer potent antioxidant capabilities, enabling curcumin to act as an effective electron donor that neutralizes reactive oxygen species (ROS) through multiple redox reactions [13,14]. Recent studies have indicated that curcumin notably improves follicular development and alleviates oocyte maturation defects in mouse models of immune-induced ovarian insufficiency, thereby preserving ovarian reserve [15]. Furthermore, curcumin nanoparticles have been shown to markedly increase the number of preantral and antral follicles in the ovaries of Rattus norvegicus exposed to carbon black. However, the precise molecular mechanisms underlying curcumin’s effects on follicular development remain unclear.

In this study, bovine ovarian granulosa cells served as the experimental model, and curcumin-treated and untreated groups were established. By integrating metabolomic and transcriptomic analyses, we aimed to systematically characterize the alterations in cellular metabolism and gene expression induced by curcumin treatment, thereby uncovering its potential molecular mechanisms in regulating follicular development.

## 2. Results

### 2.1. Effect of Curcumin Treatment on the Metabolomics of GCs in Positive Mode

To investigate the molecular mechanisms underlying curcumin’s effects on GCs, we performed untargeted metabolomic profiling following curcumin treatment. In positive ion mode, a total of 503 metabolites were significantly altered compared to the control group, with 329 upregulated and 174 downregulated (Figure 1A, Appendix A). Hierarchical clustering analysis demonstrated a clear separation between the curcumin-treated and control groups (Figure 1B). The most significantly upregulated metabolites included 4-hydroxy-4-methylglutamate, quercetagetin 3,7,3′,4′-tetramethyl ether, and laurate, while cerivastatin, 3-O-demethylfortimicin A, and capecitabine were among the most downregulated (Figure 1C). Pathway enrichment analysis revealed that these differential metabolites were primarily associated with glutathione metabolism (e.g., glutathionylspermine, glutathionylaminopropylcadaverine), chemical carcinogenesis–reactive oxygen species pathways (e.g., oxidized glutathione, glutathione), and fatty acid biosynthesis (e.g., octadecanoic acid, octanoic acid) (Figure 1D).

### 2.2. Effect of Curcumin Treatment on the Metabolomics of GCs in Negative Mode

In negative ion mode, a total of 200 metabolites were significantly altered compared to the control group, with 102 upregulated and 98 downregulated (Figure 2A, Appendix A). Hierarchical clustering analysis demonstrated a clear separation between the curcumin-treated and control groups (Figure 2B). The most significantly upregulated metabolites included pulmatin, tolmetin glucuronide, and stizolobate, while alanylglutamic acid, 4-Acetamido-2-aminobutanoic acid, and serylisoleucine were among the most downregulated (Figure 2C). Pathway enrichment analysis revealed that these differential metabolites were mainly involved in the phosphatidylinositol signaling system (e.g., 1D-myo-inositol 1,3,4,5-tetrakisphosphate), pentose phosphate pathway (e.g., sedoheptulose, D-erythrose 4-phosphate), and cholesterol metabolism (e.g., Taurochenodeoxycholate) (Figure 2D).

### 2.3. Transcriptomic Analysis of Curcumin-Treated GCs

To further elucidate the molecular mechanisms by which curcumin modulates GC function, we performed transcriptomic profiling following curcumin treatment. RNA sequencing revealed 1168 differentially expressed genes (DEGs) between the control and curcumin-treated groups, including 582 upregulated and 586 downregulated transcripts (Figure 3A, Appendix A). Hierarchical clustering demonstrated clear segregation of DEGs, indicating consistent transcriptional responses to curcumin (Figure 3B). Gene Ontology (GO) enrichment analysis showed that these DEGs were predominantly associated with cellular processes, metabolic processes, and biological regulation (Figure 3C, Appendix A). KEGG pathway analysis further revealed significant enrichment in glutathione metabolism, cellular senescence, and the Hippo signaling pathway, suggesting that curcumin modulates key metabolic and regulatory pathways in follicular granulosa cells (Figure 3D, Appendix A).

### 2.4. Integrated Transcriptomic and Metabolomic Analysis of Curcumin-Treated GCs

To further elucidate the systemic mechanisms by which curcumin regulates granulosa cell function, we integrated metabolomic and transcriptomic data and performed joint pathway enrichment analysis. The results revealed 90 commonly enriched pathways between the positive ion mode and transcriptome data (Figure 4A) and 45 shared pathways between the negative ion mode and transcriptome data (Figure 4B). Among them, the top 30 significantly enriched pathways in the positive ion mode included glutathione metabolism, AMPK signaling, FoxO signaling, glycolysis/gluconeogenesis, cysteine and methionine metabolism, apoptosis, and necroptosis (Figure 4C). In contrast, the top 30 pathways enriched in the negative ion mode included bile secretion, fatty acid metabolism, phosphatidylinositol signaling, and ferroptosis (Figure 4D).

## 3. Discussion

Curcumin is a natural polyphenolic compound with diverse biological activities. It has recently demonstrated significant potential in regulating ovarian function and follicular development [16,17,18]. Previous studies have shown that curcumin can alleviate follicular dysregulation in models of diminished ovarian reserve (DOR), polycystic ovary syndrome (PCOS), and premature ovarian insufficiency (POI), primarily through its antioxidant, anti-apoptotic, and anti-inflammatory properties. In DOR models, curcumin has been reported to promote the activation of primordial follicles and enhance ovarian reserve by activating the TGF-β1/AMH and AMPK/SIRT1 signaling pathways [19,20]. In PCOS models, curcumin improves insulin sensitivity via modulation of the IRS1/PI3K/GLUT4 pathway and inhibits PTEN expression, thereby reducing ovarian inflammation and supporting normal follicular development [21]. In POI models, curcumin activates the Nrf2/HO-1 and PI3K/Akt signaling pathways, leading to decreased oxidative stress and granulosa cell apoptosis, upregulation of AMH expression, and increased follicle counts, ultimately improving the ovarian microenvironment [22]. However, the precise molecular mechanisms underlying curcumin’s effects remain to be fully elucidated.

Metabolomic analysis revealed that curcumin treatment induced significant alterations in granulosa cell metabolites. Under positive ion mode, 503 metabolites were significantly affected, including 329 upregulated and 174 downregulated compounds. In negative ion mode, 200 metabolites exhibited significant changes, with 102 upregulated and 98 downregulated. Among the most significantly upregulated metabolites in positive ion mode were 4-hydroxy-4-methylglutamate, quercetagetin 3,7,3′,4′-tetramethyl ether, and laurate, while cerivastatin, 3-O-demethylfortimicin A, and capecitabine were notably downregulated. In negative ion mode, pulmatin, Tolmetin glucuronide, and stizolobate were the most significantly upregulated metabolites, whereas alanylglutamic acid, 4-acetamido-2-aminobutanoic acid, and serylisoleucine were significantly downregulated. Notably, previous studies have shown that dietary supplementation with quercetagetin enhances antioxidant function, improves liver mitochondrial activity, and modulates gut microbiota in broilers under high stocking density. These effects are mediated via activation of the Nrf2/antioxidant response element (ARE) signaling pathway through Keap1 regulation [23,24]. Additionally, lauric acid has been reported to alleviate inflammation and mitigate structural damage in the lungs of type II diabetic rats [25]. These findings suggest that curcumin may contribute to the maintenance of granulosa cells at least in part by modulating the levels of bioactive metabolites such as quercetagetin derivatives and lauric acid, which are known for their antioxidant and anti-inflammatory properties.

Pathway enrichment analysis showed that the differentially expressed metabolites were mainly enriched in key pathways such as glutathione metabolism, fatty acid biosynthesis, and cholesterol metabolism. Notably, the significant enrichment of the glutathione metabolism pathway suggests that curcumin may enhance antioxidant capacity, maintain intracellular redox homeostasis, and thereby protect granulosa cells from ferroptosis and apoptosis. These findings are consistent with previous studies. For instance, Wang et al. reported that curcumin enhanced antioxidant capacity in Macrobrachium rosenbergii by regulating the expression of antioxidant genes such as SOD, CAT, and GPX [26]. Similarly, Yan et al. demonstrated that curcumin alleviated ovarian oxidative stress and delayed follicular atresia by activating the Nrf2/HO-1 signaling pathway [22].

A previous transcriptomic analysis investigating the effects of curcumin on dihydrotestosterone-induced ovarian granulosa cells revealed that curcumin upregulated enzymes involved in estrogen synthesis, downregulated genes associated with lipid metabolism and glucuronic acid processes, inhibited androgen receptor (AR) activity, significantly improved cell viability, and restored normal granulosa cell development. Gene set enrichment and pathway analyses further suggested curcumin’s protective role in granulosa cell function [27]. In the present study, transcriptomic analysis further identified a total of 1168 differentially expressed genes (DEGs) in response to curcumin treatment, including 582 upregulated and 586 downregulated genes. Gene Ontology (GO) enrichment analysis indicated that these DEGs were primarily involved in metabolic processes, cellular processes, and biological regulation. KEGG pathway analysis revealed that the DEGs were significantly enriched in glutathione metabolism, the Hippo signaling pathway, and cellular senescence. Notably, the Hippo pathway plays a crucial role in follicular activation and granulosa cell proliferation [28,29], aligning well with the metabolic alterations observed in the metabolomic data.

To further elucidate the systemic mechanisms underlying curcumin regulation of granulosa cell function, we performed an integrated pathway enrichment analysis by combining metabolomic and transcriptomic datasets. In the joint analysis of the positive ion mode and transcriptomic data, both differentially expressed metabolites and genes were co-enriched in the glutathione metabolism pathway, suggesting that curcumin may enhance glutathione synthesis and redox capacity to alleviate oxidative stress in granulosa cells. In addition, shared enrichment was observed in several key pathways, including the AMPK signaling pathway, FoxO signaling pathway, glycolysis/gluconeogenesis, cysteine and methionine metabolism, apoptosis, and necroptosis. AMPK is a central energy sensor that regulates antioxidant gene expression, autophagy, and cell survival. Previous studies have demonstrated that curcumin restores redox homeostasis in the ovary by activating the AMPK/SIRT1 axis [20] and protects mouse ovaries from oxidative damage via AMPK/mTOR-mediated autophagy [30]. The FoxO pathway, a critical regulator of cell cycle control and oxidative stress responses in follicular cells [31], was also significantly activated by curcumin treatment, suggesting its potential role in promoting granulosa cell survival. In our joint analysis of negative ion mode and transcriptomic data, curcumin significantly modulated pathways involved in bile secretion, fatty acid metabolism, biosynthesis of unsaturated fatty acids, phosphatidylinositol signaling, tryptophan/tyrosine metabolism, and ferroptosis. Notably, enrichment in apoptosis, necroptosis, and ferroptosis pathways was particularly prominent, aligning with accumulating evidence that these cell death pathways play pivotal roles in follicular development [8,32,33,34,35]. Thus, our findings suggest that curcumin may regulate granulosa cell fate by modulating glutathione biosynthesis and redox balance, thereby influencing apoptosis and ferroptosis.

## 4. Materials and Methods

### 4.1. Bovine Granulosa Cell Culture

Bovine ovaries were collected from healthy, non-pregnant cows at a local abattoir in Mianyang, Sichuan, China. The animals were not subjected to any experimental treatment prior to ovary collection. Ovaries were transported to the laboratory within 2 h post-slaughter. Upon arrival, ovaries were disinfected in 75% ethanol for 30 s and rinsed with sterile PBS. After removing surrounding connective tissue, follicles were punctured, and granulosa cells were collected by gently scraping the inner follicular wall with sterile forceps. The cells were suspended in DMEM/F12 basal medium 5% fetal bovine serum (FBS, Gibco, NY, USA), centrifuged at 1000 rpm for 3 min, resuspended, and seeded into T25 culture flasks. Once the cells reached 85–90% confluence, they were transferred into six-well plates and incubated for 24 h before treatment. For the curcumin treatment group, primary granulosa cells were treated with 2 μM curcumin for 24 h.

### 4.2. Library Preparation and Transcriptome Analysis

Total RNA was extracted from three control samples and three curcumin-treated samples for RNA-seq analysis. Library construction was performed by Novogene Co., Ltd. (Beijing, China) using the NEBNext^®^ Ultra™ RNA Library Prep Kit for Illumina^®^ (New England Biolabs, Ipswich, MA, USA), following the manufacturer’s protocol. mRNA was enriched from total RNA using poly-T oligo-attached magnetic beads. First-strand cDNA was synthesized using random hexamer primers and M-MuLV Reverse Transcriptase followed by second-strand synthesis using DNA Polymerase I and RNase H. The resulting cDNA fragments underwent end-repair, 3′ adenylation, and adaptor ligation. Size selection of 150–200 bp fragments was carried out using the AMPure XP system (Beckman Coulter, CA, USA), and the selected fragments were then enriched by PCR amplification. Library quality was assessed using the Agilent 2100 Bioanalyzer (Agilent Technologies, Santa Clara, CA, USA). Final libraries were sequenced on an Illumina NovaSeq platform to generate 150 bp paired-end reads.

Raw sequencing reads in FASTQ format were processed with fastp to remove adaptor sequences, poly-N regions, and low-quality reads. The reference genome index was constructed using HISAT2 v2.0.5, and clean paired-end reads were aligned to the reference genome using the same software. Gene-level read counts were obtained using featureCounts v1.5.0-p3. Differential gene expression analysis between the control and treatment groups was performed using the DESeq2 package (v1.20.0) in R. *p*-values were adjusted for multiple testing using the Benjamini–Hochberg method to control the false discovery rate. Genes with an FDR < 0.05 were considered significantly differentially expressed.

### 4.3. Metabolite Extraction

Metabolomic profiling was conducted using a Waters Acquity I-Class PLUS ultra-performance liquid chromatography (UPLC) system coupled with a Waters Xevo G2-XS QTof high-resolution mass spectrometer. Metabolite separation was performed on a Waters Acquity UPLC HSS T3 column. Both positive and negative ion modes were employed. Mobile phase A consisted of 0.1% formic acid in water, while mobile phase B consisted of 0.1% formic acid in acetonitrile. The injection volume for each sample was 2 μL.

### 4.4. LC-MS/MS Analysis

Mass spectrometric data were acquired in MSe mode using MassLynx V4.2 software, allowing simultaneous collection of precursor and product ion data within a single scan cycle. Low collision energy was disabled, while high collision energy was set in the range of 10–40 V. Each scan was performed with a duration of 0.2 s. The electrospray ionization (ESI) source parameters were as follows: capillary voltage +2500 V in positive mode and −2000 V in negative mode, cone voltage 30 V, source temperature 100 °C, desolvation temperature 500 °C, cone gas flow rate 50 L/h, and desolvation gas flow rate 800 L/h.

### 4.5. Data Processing and Annotation

Raw LC-MS/MS data files were processed using Progenesis QI software (version 4.0), which included peak picking, alignment, and normalization. Metabolite identification was achieved by matching the acquired spectra against the METLIN database and an in-house spectral library, both integrated within the Progenesis QI platform.

Normalized peak area data were subjected to multivariate and statistical analyses. Principal component analysis (PCA) and Spearman correlation analysis were used to assess sample consistency and quality control. Metabolite annotation, including compound classification and pathway mapping, was performed using the KEGG, HMDB, and LipidMaps databases [36,37]. Differential metabolites were selected based on the criteria: fold change (FC) > 1, *p*-value < 0.05, and variable importance in projection (VIP) > 1. Statistical significance was assessed using Student’s *t*-test. Orthogonal partial least squares discriminant analysis (OPLS-DA) was conducted using the “ropls” package in R, and model reliability was validated by 200 permutation tests. VIP scores were calculated through cross-validation. KEGG pathway enrichment of differential metabolites was evaluated using hypergeometric distribution testing.

## 5. Conclusions

In conclusion, our integrative transcriptomic and metabolomic analyses revealed that curcumin regulates bovine ovarian granulosa cell function by modulating key metabolic and signaling pathways. Notably, curcumin significantly affected glutathione metabolism and pathways related to apoptosis, necroptosis, and ferroptosis. These findings indicate that curcumin may support follicular development by enhancing antioxidant defenses and regulating programmed cell death, providing novel insights into its potential therapeutic role in improving ovarian function.

## Figures and Tables

**Figure 1 ijms-26-06713-f001:**
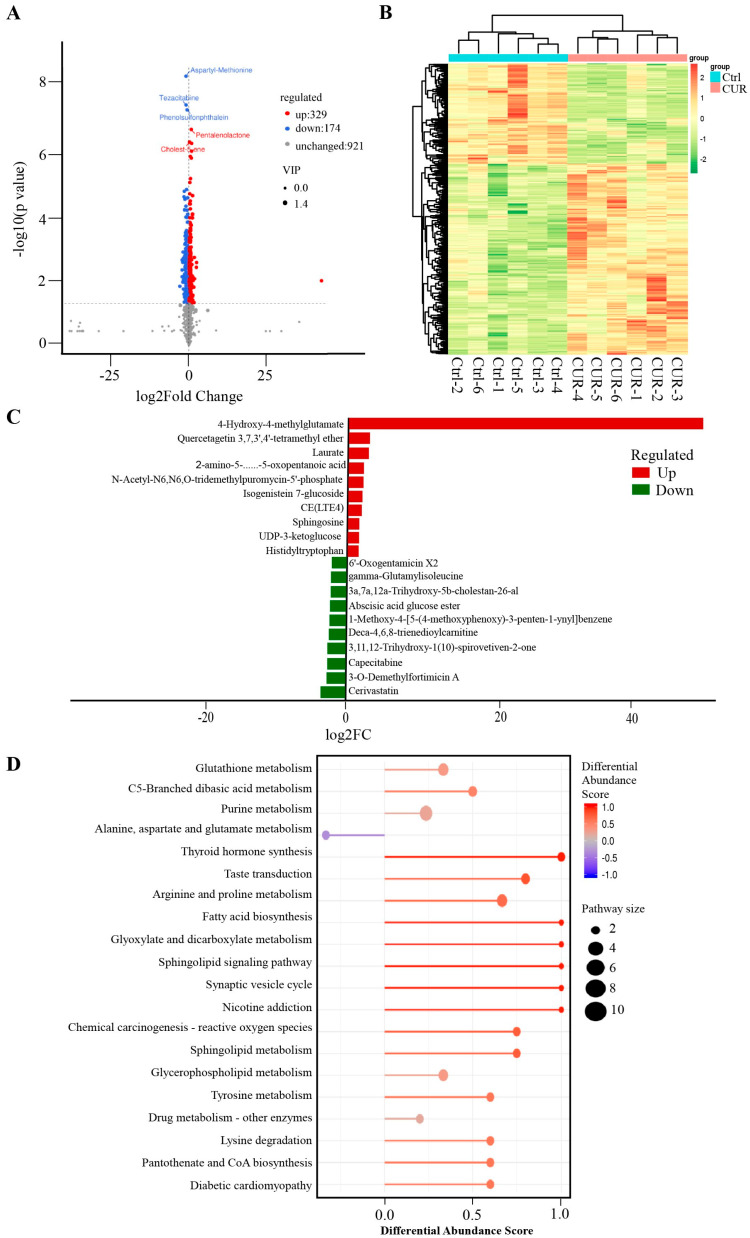
Positive Mode Metabolomics of Curcumin-Treated GCs: (**A**) Volcano of metabolites with significant differences between control and curcumin-treated groups. The criteria for upregulated differential metabolites should be VIP > 1, *p* < 0.05, and log2(Foldchange) > 0, while the criteria for downregulated differential metabolites should be the VIP > 1, *p* < 0.05, and log2(Foldchange) < 0. (**B**) Clustering heat map of differential metabolites between control and curcumin-treated groups. (**C**) Top 10 elevated and reduced differential metabolites identified between the control and curcumin-treated groups in the positive model (**D**) Kyoto Encyclopedia of Genes and Genomes (KEGG) enriched by differential metabolites between control and curcumin-treated groups.

**Figure 2 ijms-26-06713-f002:**
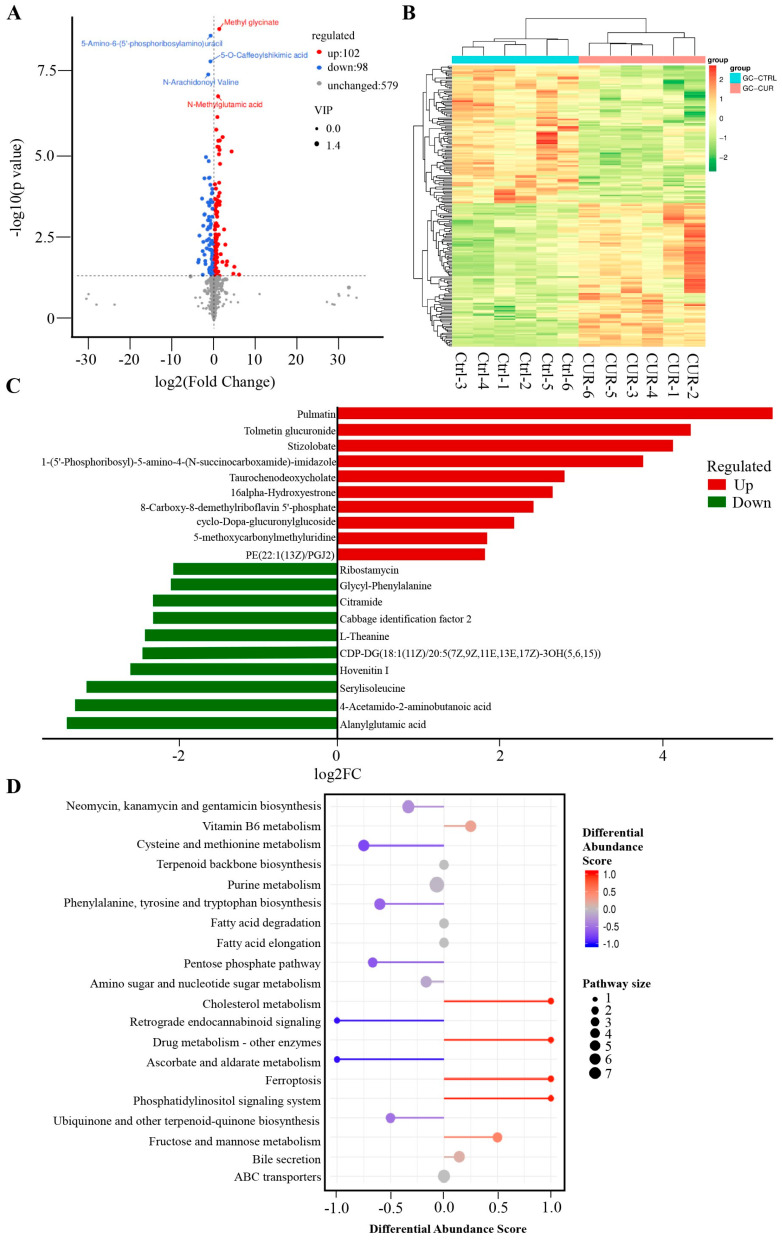
Negative Mode Metabolomics of Curcumin-Treated GCs: (**A**) Volcano of metabolites with significant differences between control and curcumin-treated groups. The criteria for upregulated differential metabolites should be VIP > 1, *p* < 0.05, and log2(Foldchange) > 0, while the criteria for downregulated differential metabolites should be VIP > 1, *p* < 0.05, and log2(Foldchange) < 0. (**B**) Clustering heat map of differential metabolites between control and curcumin-treated groups. (**C**) Top 10 elevated and reduced differential metabolites identified between the control and curcumin-treated groups in the positive model (**D**) KEGG pathways enriched by differential metabolites between control and curcumin-treated groups.

**Figure 3 ijms-26-06713-f003:**
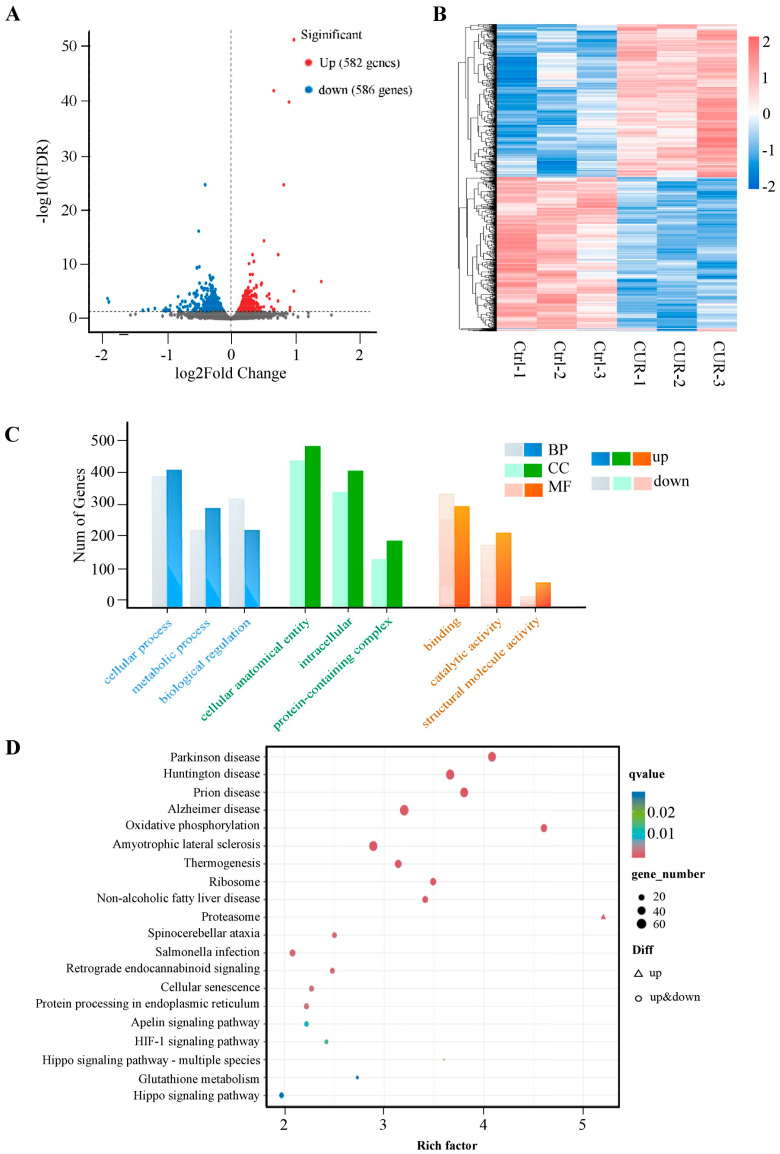
Analysis of differentially expressed genes: (**A**) Volcano plots of the differential genes between control and curcumin-treated groups, genes with FDR < 0.05 were considered significantly differentially expressed (colored in blue or red), while grey dots represent genes with FDR ≥ 0.05; (**B**) Clustering heat map of differential metabolites between control and curcumin-treated groups; (**C**) Gene ontology (GO) analysis of part of differential genes. (**D**) KEGG pathway analysis of part of differential genes.

**Figure 4 ijms-26-06713-f004:**
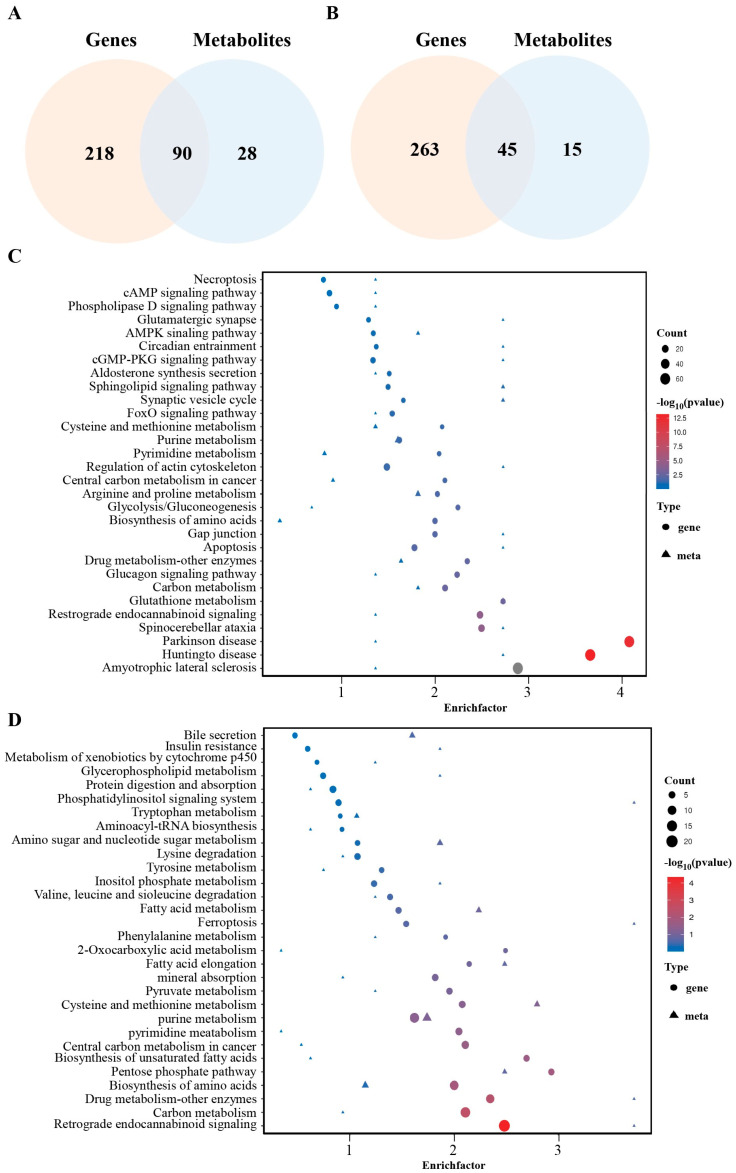
Joint pathway enrichment analysis based on integrated metabolomic and transcriptomic data. (**A**) Venn diagram showing 90 co-enriched pathways between the positive ion mode and transcriptomic data. (**B**) Venn diagram showing 45 co-enriched pathways between the negative ion mode and transcriptomic data. (**C**) Top 30 significantly enriched pathways in the positive ion mode. (**D**) Top 30 enriched pathways in the negative ion mode.

## Data Availability

The data underlying this article will be shared upon reasonable request to the corresponding author.

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
