# Peer review of "Integrated Metabolomic and Transcriptomic Analysis Reveals the Regulatory Effects of Curcumin on Bovine Ovarian Granulosa Cells"

_ijms, 2025, doi:10.3390/ijms26146713_

Round 1
Reviewer 1 Report
Comments and Suggestions for Authors
the manuscript is well design and presented, however, it needs careful English editing especially the introduction
Comments on the Quality of English Languageneeds English editing
Author Response
the manuscript is well design and presented, however, it needs careful English editing especially the introduction.
Response: We thank the reviewer for the valuable comment. In response, we have revised the manuscript thoroughly, with particular attention to the Introduction section. We have carefully restructured and refined the language for clarity and readability. All revised parts are highlighted in yellow for easy reference. Please see the updated Introduction section below:
Introduction
The ovary is a highly organized and complex organ consisting of germ cells (oocytes) and somatic cells, including granulosa cells, theca cells, and stromal cells [1]. Notably, the intimate interactions between granulosa cells and oocytes are essential for determining oocyte quality [2,3]. Granulosa cells facilitate oocyte development by supplying nutrients and metabolites through gap junction-mediated communication [4], and protect oocytes from oxidative stress through intrinsic antioxidant mechanisms [5,6], thereby ensuring proper maturation and ovulation [7]. Apoptosis and autophagy of granulosa cells are physiological processes during follicular development that help eliminate dysfunctional follicles and maintain a pool of healthy ones within the ovary [8]. However, excessive activation of these cell death pathways may lead to follicular atresia and depletion, ultimately contributing to ovarian dysfunction [9].
Curcumin is a natural lipophilic polyphenolic compound extracted from the rhizomes of Zingiberaceae plants, such as Curcuma longa and Curcuma aromatica, includes two aromatic ring systems containing o-methoxy phenolic groups, connected by a seven-carbon linker that consists of an α,β-unsaturated β-diketone moiety. This distinctive structure confers strong membrane permeability, allowing efficient cellular uptake [10]. It has been widely recognized for its extensive biological activities and pharmacological effects, including anti-inflammatory, antioxidant, and anticancer properties, making it a promising therapeutic candidate for diverse health conditions [11,12]. The conjugated double bonds in its structure confer potent antioxidant capabilities, enabling curcumin to act as an effective electron donor that neutralizes reactive oxygen species (ROS) through multiple redox reactions [13,14]. Recent studies have indicated that curcumin notably improves follicular development and alleviates oocyte maturation defects in mouse models of immune-induced ovarian insufficiency, thereby preserving ovarian reserve [15]. Furthermore, curcumin nanoparticles have been shown to markedly increase the number of preantral and antral follicles in the ovaries of Rattus norvegicus exposed to carbon black. However, the precise molecular mechanisms underlying curcumin’s effects on follicular development remain unclear.
In this study, bovine ovarian granulosa cells served as the experimental model, and curcumin-treated and untreated groups were established. By integrating metabolomic and transcriptomic analyses, we aimed to systematically characterize the alterations in cellular metabolism and gene expression induced by curcumin treatment, thereby uncovering its potential molecular mechanisms in regulating follicular development.
We appreciate the reviewer’s suggestion, which has helped us improve the overall quality and presentation of the manuscript.

Reviewer 2 Report
Comments and Suggestions for Authors
Comments to the manuscript “Integrated Metabolomic and Transcriptomic Analysis Reveals the Regulatory Effects of Curcumin on Bovine Ovarian Granulosa Cells”
The manuscript is original and well-written. Molecular mechanisms underlying curcumin's effects on bovine ovarian granulosa were revealed through integrated metabolomic and transcriptomic analyses. Significant new information is provided. The objectives and the rationale of the study are clearly stated and the conclusions are supported by the results.
Minor comments.
-Line 48, please briefly describe the chemical structure of cucurmin.
-Fig 1C, line 77: “Pathway enrichment analysis revealed that these differential metabolites were mainly involved in glutathione metabolism, chemical carcinogenesis–reactive oxygen species, and fatty acid biosynthesis pathways”. It could be more precise to indicate that the differential metabolites were involved in these three processes, among others. The same applies for figure 2D, line 99.
-Fig 3A, please include the number of upregulated and downregulated transcripts.
-The discussion could be enriched whether results from the metabolomic assay are analyzed. What could be the role of the most significantly up- and down-regulated metabolites in the protective role of curcumin? (for instance, quercetagetin 3,7,3',4'-tetramethyl ether, laurate, cerivastatin, 3-O-demethylfortimicin A, capecitabine). In addition, please discuss results of the following papers:
Chen D, Yu Q, Sheng S, Cai L, Zheng J, Zhang Y. Transcriptomic analysis of the effects exerted by curcumin on dihydrotestosterone-induced ovarian granulosa cells. Front Endocrinol (Lausanne). 2025 Feb 13;16:1522269. doi: 10.3389/fendo.2025
Duan H, Yang S, Yang S, Zeng J, Yan Z, Zhang L, Ma X, Dong W, Zhang Y, Zhao X, Hu J, Xiao L. The mechanism of curcumin to protect mouse ovaries from oxidative damage by regulating AMPK/mTOR mediated autophagy. Phytomedicine. 2024 Jun;128:155468. doi: 10.1016/j.phymed.2024
Author Response
The manuscript is original and well-written. Molecular mechanisms underlying curcumin's effects on bovine ovarian granulosa were revealed through integrated metabolomic and transcriptomic analyses. Significant new information is provided. The objectives and the rationale of the study are clearly stated and the conclusions are supported by the results.
Minor comments.
-Line 48, please briefly describe the chemical structure of cucurmin.
Response: We thank the reviewer’s comment. As suggested, we have added a brief description of the chemical structure of curcumin in Lines 51–52: “Curcumin … includes two aromatic ring systems containing o-methoxy phenolic groups, connected by a seven-carbon linker that consists of an α,β-unsaturated β-diketone moiety.”
-Fig 1C, line 77: “Pathway enrichment analysis revealed that these differential metabolites were mainly involved in glutathione metabolism, chemical carcinogenesis–reactive oxygen species, and fatty acid biosynthesis pathways”. It could be more precise to indicate that the differential metabolites were involved in these three processes, among others. The same applies for figure 2D, line 99.
Response: We thank the reviewer’s comment. As suggested, we have revised the statements to clarify that the differential metabolites were involved in these three pathways among others. Please refer to Lines 82–86 and 105–109 for the updated descriptions
-Fig 3A, please include the number of upregulated and downregulated transcripts.
Response:We thank the reviewer’s suggestion. As requested, we have added the number of upregulated and downregulated transcripts in the updated Figure 3A. Please refer to the revised figure in the latest version of the manuscript.
-The discussion could be enriched whether results from the metabolomic assay are analyzed. What could be the role of the most significantly up- and down-regulated metabolites in the protective role of curcumin? (for instance, quercetagetin 3,7,3',4'-tetramethyl ether, laurate, cerivastatin, 3-O-demethylfortimicin A, capecitabine). In addition, please discuss results of the following papers:
Chen D, Yu Q, Sheng S, Cai L, Zheng J, Zhang Y. Transcriptomic analysis of the effects exerted by curcumin on dihydrotestosterone-induced ovarian granulosa cells. Front Endocrinol (Lausanne). 2025 Feb 13;16:1522269. doi: 10.3389/fendo.2025
Duan H, Yang S, Yang S, Zeng J, Yan Z, Zhang L, Ma X, Dong W, Zhang Y, Zhao X, Hu J, Xiao L. The mechanism of curcumin to protect mouse ovaries from oxidative damage by regulating AMPK/mTOR mediated autophagy. Phytomedicine. 2024 Jun;128:155468. doi: 10.1016/j.phymed.2024
Response: We thank the reviewer for this valuable suggestion. As requested, we have expanded the discussion to address the potential roles of the most significantly up- and down-regulated metabolites in the protective effects of curcumin. In particular, we discussed the antioxidant functions of quercetagetin and laurate (Lines 178–197). Additionally, we have incorporated discussion of the following relevant studies: the work by Chen et al. (2025), which demonstrated that curcumin promotes granulosa cell development by modulating estrogen synthesis and suppressing androgen receptor activity (Lines 209–215), and the study by Duan et al. (2024), which showed that curcumin protects mouse ovaries from oxidative damage through the AMPK/mTOR-mediated autophagy pathway (Lines 237–238), supporting the mechanistic observations in our study. These revisions have been incorporated into the revised manuscript as indicated. We sincerely appreciate the reviewer’s insightful comments.

Reviewer 3 Report
Comments and Suggestions for Authors
Review of the following manuscript:
Manuscript ID: ijms-3662872
Title: Integrated Metabolomic and Transcriptomic Analysis Reveals the
Regulatory Effects of Curcumin on Bovine Ovarian Granulosa Cells
Authors: Bingfei Zhang, Le Chen, Liping Mei, Xianbo Jia, Shiyi Chen, Jie
Wang, Hengwei Yu, Songjia Lai, Wenqiang Sun *
In the article submitted to me for review, the authors investigated the regulatory effects of curcumin on bovine ovarian granulosa cells by integrated transcriptomic and metabolomic analyses. Multi-omics analysis further showed that curcumin significantly affects pathways related to glutathione metabolism, cysteine and methionine metabolism, and multiple forms of programmed cell death, including apoptosis, necroptosis, and ferroptosis.
- The generalization made that curcumin may exert its beneficial effect on follicle development should be redacted as an assumption since the authors did not handle oocytes. The granulosa cell-oocyte loop is narrow and specific, but in no way does it guarantee that the resupply of granulosa cells that respond to the same pathways in oocytes. On the contrary, CoCs work in a suspicious, complementary manner (L28-29). The conclusion very well noted " may support".
- The introduction adequately describes the problem and emphasizes why the issue is important. A specific purpose of the study is stated.
- The description of methods, cell cultures, does not make it clear where these ovaries came from to exclude their treatment and their relevance to the results obtained. How many are there? What cell yield are you doing is also not clear?
- In the library, you start with three controls plus three cucumicin-treated? This is too insufficient for transcriptome analysis. You can't check the static bundles that are specified when n = ? , is the variance equal between the experimental and control?
- Note: You talk about collected granulosa cells in the methods, and among this in the results and link to discussion, total follicular cells are mentioned. However, follicular cells are the aggregate of granulosa plus the most intimate to the oocyte (cumulus cells). When you describe in the first step how you are collecting your samples you will be able to judge what you are working with because the genomics and metabolomics of theca, of granulosae, of cumulus are different. Important to distinguish as they have different signals.
- The results are presented in 4 figures, but are referred to twice as Figure 2 in the figure captions, which does not correspond with the text;
- Ethical norms corresponding to the research performed are indicated.
- There is no conflict of interest in the collective.
- Thirty-two sources are cited, with a large percentage from the last five years;
` In conclusion, minor technical errors were found. I recommend that it be revised.
Accepted after minor revision.
Author Response
We would like to thank the reviewer for their time and effort.
